

SciPost Phys. Comm. Rep. 15 (2026)

# Simplified template cross sections – Stage 1.1 and 1.2

Nicolas Berger[1], Claudia Bertella[2,3], Matteo Bonanomi[4], Nihal Brahimi[1],
Thomas P. Calvet[5], Milene Calvetti[6], Valerio Dao[2], Marco Delmastro[1],
Michael Duehrssen-Debling[2], Paolo Francavilla[6], Yacine Haddad[7], Sarah Heim[8,4],
Jelena Jovicevic[9], Oleh Kivernyk[1], Maria Moreno Llacer[10], Jonathon M. Langford[11],
Changqiao Li[12], Giovanni Marchiori[13], Josh A. McFayden[14], Johannes K. L. Michel[15],
Predrag Milenovic[2,16], Carlo E. Pandini[2], Edward Scott[11], Frank J. Tackmann[8],
Kerstin Tackmann[8,4], Lorenzo Viliani[17], Meng Xiao[18], and Hongtao Yang[12]

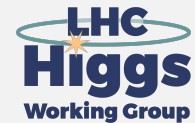

*Part of the Regular Reports Collection*
*published in the LHC Higgs Working Group Reports Series*

## Abstract

Simplified Template Cross Sections (STXS) have been adopted by the LHC experiments as a common framework for Higgs measurements. Their purpose is to reduce the theoretical uncertainties that are directly folded into the measurements as much as possible, while at the same time allowing for the combination of the measurements between different decay channels as well as between experiments. We report the complete, revised definition of the STXS kinematic bins (stage 1.1 and stage 1.2), which have been used for the measurements by the ATLAS and CMS experiments using the full LHC Run 2 datasets. The main focus is on the four dominant Higgs production processes, namely gluon-fusion, vector-boson fusion, production in association with a vector boson and in association with a $t\bar{t}$ pair. We also comment briefly on the treatment of other production modes.

| | |
|---|---|
| Received | 2025-07-11 |
| Accepted | 2025-12-04 |
| Published | 2026-01-07 |

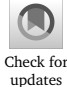

**1** LAPP, Annecy, France
**2** CERN, Geneva, Switzerland
**3** Institute of High Energy Physics, Chinese Academy of Sciences, Beijing, China
**4** University of Hamburg, Hamburg, Germany
**5** Stony Brook University, Stony Brook, NY, USA
**6** Università and INFN Sezione di Pisa, Pisa, Italy
**7** Northeastern University, Boston, MA, USA
**8** Deutsches Elektronen-Synchrotron (DESY), Hamburg, Germany
**9** Institute of Physics Belgrade, Belgrade, Serbia
**10** University of Valencia and CSIC, Spain
**11** Imperial College, London, United Kingdom
**12** University of Science and Technology of China, Hefei, China
**13** LPNHE, Sorbonne Université, Paris Diderot Sorbonne Paris Cité,
CNRS/IN2P3, Paris, France

**14** University of Sussex, Brighton, United Kingdom
**15** University of Amsterdam, Amsterdam, Netherlands
**16** University of Belgrade, Belgrade, Serbia
**17** Università and INFN Sezione di Firenze, Firenze, Italy
**18** Johns Hopkins University, Baltimore, MD, USA

## Contents

## 1 Introduction

Simplified Template Cross Sections (STXS) have been adopted by the LHC experiments as an evolution of the signal strength measurements performed during Run 1 of the LHC. They were first discussed in detail in Section III.3 of [1] and Section III.2 of [2]. Their purpose is twofold. They provide more fine-grained measurements for individual Higgs production modes in various kinematic regions, and reduce the theoretical uncertainties that are directly folded into the measurements. At the same time, they allow for the use of multivariate analysis techniques and provide a common framework for the combination of measurements in different decay channels and eventually between experiments. Currently, STXS measurements are available in all five major Higgs decay channels, namely $H \rightarrow \gamma\gamma$ [3,4], $H \rightarrow ZZ^* \rightarrow 4\ell$ [5,6], $H \rightarrow WW^* \rightarrow 2\ell2\nu$ [7,8], $H \rightarrow \tau\tau$ [9,10], and $H \rightarrow b\bar{b}$ [11–13] (only shortly after its discovery [14,15]), as well as from the combination of several decay channels [16–20]. Both

individual and combined STXS measurements can then be used as inputs for subsequent interpretations in and beyond the Standard Model (SM). This can be the determination of overall signal strengths or coupling scale factors, SMEFT coefficients, or tests of specific BSM models, see for example [21–25].

After the first successful STXS measurements and the experience gained from them, several refinements to the definitions of the kinematic STXS bins given in [1, 2] (henceforth referred to as stage 1.0) were necessary. This paper provides the complete and revised definitions of the STXS bins, referred to as stage 1.1 and its subsequent further refinement stage 1.2. They are the result of many fruitful discussions and dedicated studies by members of the ATLAS and CMS experiments and the theory community. The STXS stages 1.1 and 1.2 presented here have been agreed upon in the context of the LHC Higgs Cross Section Working Group, and they have been used as the baseline for the measurements based on the full Run 2 datasets by ATLAS and CMS.

As discussed in more detail in Section 4.2, the vector-boson fusion (VBF) process in particular required a substantial reorganization compared to the previous stage 1.0 to be able to better exploit the potential improvements in the full Run 2 measurements for this process. For this reason, the changes are also not backward compatible with the previous stage 1.0, in the sense that they do not just correspond to a splitting of the previously defined bins. This also lead to corresponding changes in the VBF-like bins of the gluon-fusion ($gg \to H$) process. All other refinements for the $gg \to H$ and $VH$ processes are backward-compatible with stage 1.0.

The remainder of this paper is organized as follows. In Section 2, we briefly review the main features and goals of the STXS framework. In Section 3, we summarize the truth definitions of the relevant final-state objects, namely leptons, jets, and the Higgs boson itself. In Section 4, we give the complete bin definitions for $gg \to H$ (Section 4.1), electroweak $qqH$ production (Section 4.2), leptonic $VH$ production (Section 4.3), and $t\bar{t}H$ production (Section 4.4). In Section 4.5, we briefly comment on the current treatment of $b\bar{b} \to H$ and $tH$ production. We conclude in Section 5.

## 2 Overview

The STXS are physical cross sections (in contrast to e.g. signal strengths). They are defined in mutually exclusive regions of phase space ("bins"). Their primary features and design goals are briefly reviewed in the following.

First, the kinematic cuts defining the bins are abstracted and simplified compared to the exact fiducial volumes of the individual analyses in different Higgs decay channels. In particular, the STXS are defined inclusively in the Higgs boson decay (up to an overall cut on the rapidity of the Higgs boson). The measurements are unfolded to the STXS bins, which are common for all analyses. This is the key feature that allows for a subsequent global combination of all measurements in different decay channels as well as from ATLAS and CMS. When combining measurements in different decay channels, one can either assume the SM branching ratios or consider the ratios of the branching ratios as additional free parameters.

While being simplified to allow for the combination of different measurements, the bin definitions nevertheless try to be as close as possible to the typical experimental kinematic selections or more generally the kinematic regions that dominate the experimental sensitivity. The goal is to allow for the use of advanced analysis techniques such as event categorization or multivariate techniques in order to achieve maximal sensitivity, while still keeping the unfolding uncertainties small. In particular, an important goal is to avoid any unnecessary extrapolations and as much as possible reduce the dependence on theory predictions and uncertainties that are folded into the measurements.

The second key feature of STXS is that they are defined for specific production modes, with the SM production processes serving as kinematic templates. This separation into production modes is an essential aspect to reduce the model dependence, i.e., to eliminate the dependence of the measurements on the relative fractions of the production modes in the SM.

From the above discussion it should be clear that STXS measurements should not replace measurements of fully fiducial and differential cross sections in individual decay channels. Rather, they complement each other and are optimized for somewhat different purposes. In particular, the STXS allow testing the SM in the kinematics of the different Higgs production modes with an improved sensitivity from combining all decay channels.

For the concrete definitions of the STXS bins, several considerations have to be taken into account. The key goals are to

- minimize the dependence on theory uncertainties that are folded into the measurements,
- maximize the experimental sensitivity,
- isolate possible BSM effects,
- and limit the number of bins to match the experimental sensitivity.

The last point in particular deserves to be stressed, as it is an important practical consideration. It is often in direct competition with the other requirements, and so they must be balanced against each other. In practice, for an analysis to contribute to the global combination, it needs to implement the complete split at the truth level, even if it only measures a small subset of bins. Therefore, keeping the number of bins at a manageable level is essential to facilitate the practical implementation and keep the required overhead manageable for all analyses. In addition, it reduces the technical complications that arise when one has to statistically combine many weakly constrained or unconstrained measurements.

The number of separately measured bins can evolve with time, such that the measurements can become more fine-grained as the size of the available dataset increases. For this purpose, different stages are defined, corresponding to increasingly fine-grained measurements. The stage 0 bin definitions essentially correspond to the production mode measurements of Run 1. The stages 1.1 and 1.2 reported here update the original stage 1.0, and target the full Run 2 measurements. Compared to stage 1.1, in stage 1.2 the $gg \to H$ binning has been extended and a binning for the $t\bar{t}H$ process has been added. It should be stressed that the goal is not that the complete set of bins should be measurable by any single analysis, but rather that the full granularity should become accessible in the combination of all decay channels with the full Run 2 dataset. In individual analyses several bins can be merged and only their sum be measured according to the sensitivity of each analysis and decay channel.

## 2.1 Sub-bin boundaries for theory uncertainties

One important goal is to reduce the theory dependence of the measurements. First, this requires avoiding that the measurements have to extrapolate from a certain measured region in phase space to a much larger region of phase space, in particular when such an extrapolation entails nontrivial theory uncertainties. More generally, it requires avoiding cases with a large variation in the experimental acceptance or sensitivity within a given bin, as this introduces a direct dependence on the theory predictions for the kinematic distribution of the signal within that bin. Ideally, if such a residual theory dependence becomes a relevant source of uncertainty, the bin in question can be split further into two or more smaller bins, which moves this theory dependence on the signal distribution from the measurement into the interpretation step.

However, within many experimental analyses, the theory uncertainties on the predictions of the STXS bins, which by default should only enter in the interpretation step, do explicitly reenter the measurements whenever two bins have to be merged, e.g., due to limited statistics

or separation power. For this reason, in practice, a common treatment of theory uncertainties for all bins is important already for the measurements, even if just to know where the uncertainties are and whether two bins can be safely merged if needed or whether they should be kept split if at all possible. The detailed treatment of theory uncertainties is beyond the scope of this work and will be discussed in a separate document in preparation.

However, it is important to realize that the same basic issue also arises for the residual theory uncertainties on the signal distribution within a bin. To test and account for this dependence, essentially the same theoretical guidance is needed. For this purpose, stages 1.1 and 1.2 introduce additional sub-bin boundaries. They are meant for tracking a potential dominant source of residual theory uncertainties within a given bin. They can be viewed as potential future boundaries where a bin could be split if it becomes necessary. Defining the sub-bin boundaries already at this stage has several advantages. First, it allows for a smoother evolution of the binning, since the experimental and theoretical implementation for the new bin will already be in place in case it gets split. Secondly, it puts the treatment of the residual theory uncertainties within a not-yet split bin on a common footing with the treatment of the explicit theory uncertainties that enter in the merging of two bins. Overall, this makes the framework more robust since after all the distinction between these two cases is ultimately a matter of convention.

## 3 Definition of final-state objects

Usually, the measured event categories in all decay channels are unfolded by a fit to the STXS bins. For this purpose, and for the comparison between the measured bins and theoretical predictions from either analytic calculations or Monte Carlo (MC) simulations, the truth final state particles need to be defined unambiguously. The definition of the final-state objects, namely leptons, jets, and in particular the Higgs boson itself, are explicitly kept simpler and more idealized than in the fiducial cross section measurements. Treating the Higgs boson as on-shell final-state particle is what allows for the combination of the different decay channels.

### 3.1 Higgs boson

The STXS are defined for the production of an on-shell Higgs boson, and the unfolding should be done accordingly. A global cut on the Higgs rapidity at $|Y_H| < 2.5$ is included in all bins. As the current measurements have no sensitivity beyond this rapidity range, this part of phase space would only be extrapolated by the MC simulation. On the other hand, it is in principle possible to use electrons at very forward rapidities (up to $|\eta| \sim 5$) in $H \to ZZ^* \to 4\ell$ and extend the accessible rapidity range. For this purpose, an additional otherwise inclusive bin for $|Y_H| > 2.5$ can be included for each production process. This forward bin is not explicitly included in the following.

### 3.2 Leptons and leptonically decaying vector bosons

Leptonically decaying vector bosons, e.g. from $VH$ production, are defined from the sum of all their leptonic decay products including neutrinos. Electrons and muons from such vector-boson decays are defined as dressed, i.e., all FSR photons should be added back to the electron or muon. There should be no restriction on the transverse momentum or the rapidity of the leptons. That is, for a leptonically decaying vector boson the full decay phase space is included. Similarly, if leptonic decays to $\tau$ leptons are considered, the $\tau$ is defined from the sum of all its decay products for any $\tau$ decay mode.

### 3.3 Jets

Truth jets are defined as anti-$k_T$ jets with a jet radius of $R = 0.4$, and are built from all stable particles, including neutrinos, photons, and leptons from hadron decays or produced in the shower. Stable particles here have the usual definition, having a lifetime greater than 10 ps, i.e., those particles that are passed to GEANT4 in the experimental simulation chain.

All decay products from the Higgs boson decay are removed from the inputs to the jet algorithm, as they are accounted for by the truth Higgs boson. Similarly, all decay products from leptonic decays of signal $V$ bosons are removed, as they are treated separately. In contrast, the decay products from hadronically decaying signal $V$ bosons are included in the inputs to the truth jet building.

By default, truth jets are defined without restriction on their rapidity. A possible cut on the jet rapidity can be included in the bin definition. Unless otherwise specified, a common $p_T = 30\,\text{GeV}$ threshold for jets is used for all truth jets. In principle, a lower threshold would have the advantage to split the events more evenly between the different jet bins. Experimentally, a higher threshold at 30 GeV is favored to suppress jets from pile-up interactions, and is therefore used for the jet definition to limit the amount of extrapolation in the measurements.

## 4 Bin definitions

In this section, we give the explicit definitions of the stage 1.1 and 1.2 bins, where stage 1.2 provides an extension of stage 1.1. The sub-bin boundaries as discussed in Section 2.1 are included in the definitions and are indicated by dashed lines in the diagrams.

### 4.1 Gluon-fusion Higgs production ($gg \rightarrow H$)

The gluon-fusion template process is defined in the usual way based on the Born $gg \rightarrow H$ process plus higher-order QCD and electroweak corrections. Typically, calculations only include the virtual electroweak corrections to the Born $gg \rightarrow H$ process. We stress that here it also includes real electroweak radiation, so in particular the $gg \rightarrow Z(\rightarrow q\bar{q})H$ process.

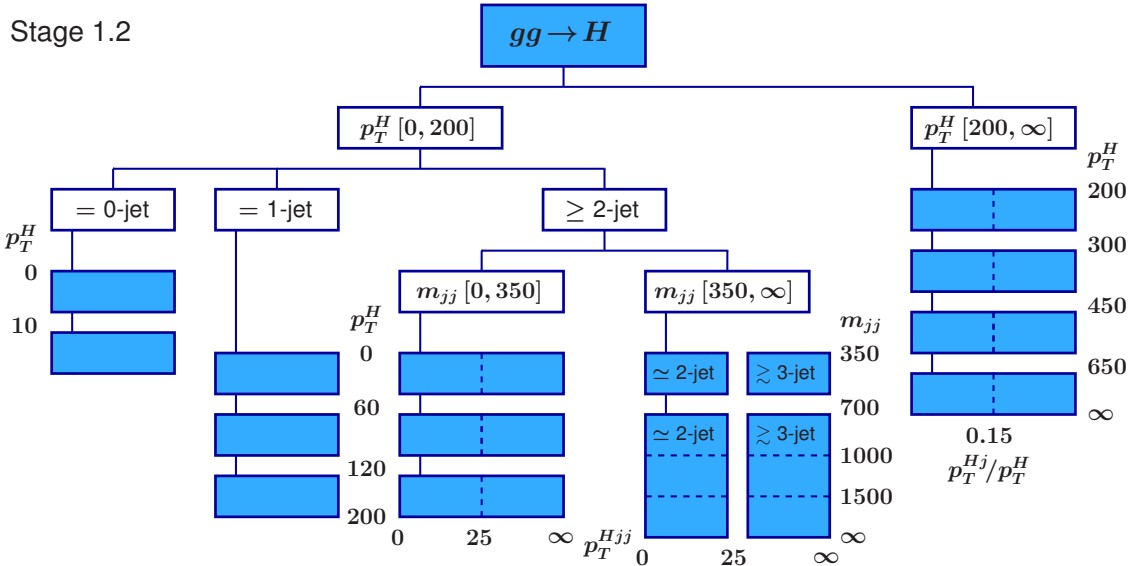

Figure 1: Stage 1.2 bins for gluon-fusion Higgs production $gg \rightarrow H$. In stage 1.1 all bins at $p_T^H > 200\,\text{GeV}$ were merged into a single bin.

The changes with respect to the previous stage 1.0 are primarily in the treatment of the BSM sensitive high-$p_T$ region, which is now split out directly as the first cut, and in a modified $N_j \geq 2$ selection matching the changes for the VBF production (see Section 4.2). Although the selection order has changed with respect to stage 1.0, the bins that describe the bulk of the $gg \rightarrow H$ production are unchanged.

The stage 1.2 bins are depicted in Figure 1 and are described briefly in the following:

- The cross section is first split into $\boxed{p_T^H < 200\,\text{GeV}}$ and $\boxed{p_T^H > 200\,\text{GeV}}$ bins. The high-$p_T^H$ region is split out first now to better enable its dedicated treatment.

  - The $\boxed{p_T^H > 200\,\text{GeV}}$ bin is primarily sensitive to BSM effects. In stage 1.0, it was part of the 1-jet and $\geq$ 2-jet bins, but in most experimental analyses it is actually merged across jet bins. This was a single bin with $p_T^H > 200\,\text{GeV}$ in stage 1.1, while in stage 1.2 it is further split into four bins, using $p_T^H = 300, 400, 650\,\text{GeV}$ as bin boundaries, to increase the sensitivity to the tails of the $p_T^H$ distribution, which can be probed by dedicated boosted analyses.
  An additional sub-bin at $p_T^{Hj}/p_T^H = 0.15$ is also introduced to more evenly divide the cross section, and to avoid increasingly large theory uncertainties that would otherwise arise from the large scale separation between a fixed jet $p_T$ threshold and the hard scale set by $p_T^H$.

  - The $\boxed{p_T^H < 200\,\text{GeV}}$ bin contains most of the cross section and is the starting point for the remaining binning.

- The $\boxed{p_T^H < 200\,\text{GeV}}$ bin is split into $\boxed{\text{0-jet}}$, $\boxed{\text{1-jet}}$, and $\boxed{\geq \text{2-jet}}$ bins, similarly to the stage 1.0 splitting.

  - Compared to stage 1.0, the $\boxed{\text{0-jet}}$ bin is split into two $p_T^H$ bins with a boundary at $p_T^H = 10\,\text{GeV}$ to probe the very low $p_T$ region of Higgs production, which contains a sizeable fraction of the cross section.

  - The $\boxed{\text{1-jet}}$ bin is split into 3 $p_T^H$ bins with boundaries at $p_T^H = 60$ and $120\,\text{GeV}$, which are unchanged with respect to stage 1.0.

  - The $\boxed{\geq \text{2-jet}}$ bin is slightly reorganized with a more dedicated split into low-$m_{jj}$ and high-$m_{jj}$ regions.

- The $\boxed{\geq \text{2-jet}}$ bin is split into low-$m_{jj}$ and high-$m_{jj}$ bins with $\boxed{m_{jj} < 350\,\text{GeV}}$ and $\boxed{m_{jj} > 350\,\text{GeV}}$, following the analogous cuts in the VBF bins. In stage 1.0, the analogous separation was implicit and it has now been made explicit. (As for the VBF bins described in Section 4.2, the $m_{jj}$ cut has been lowered from $400\,\text{GeV}$ to $350\,\text{GeV}$ and the $|\Delta\eta_{jj}|$ cut has been dropped.)

  In addition, a bin boundary is defined at $p_T^{Hjj} = 25\,\text{GeV}$, which provides a separation into 2-jet like and $\geq$ 3-jet like phase-space regions to facilitate the uncertainty treatment for $gg \rightarrow H$ as background to VBF.

  - The $\boxed{m_{jj} < 350\,\text{GeV}}$ bin contains the bulk of the $\geq$ 2-jet region. It is further split into 3 $p_T^H$ bins with boundaries at $p_T^H = 60$ and $120\,\text{GeV}$, aligned with the 1-jet bin.

This allows for an almost inclusive measurement of the $gg \to H$ $p_T^H$ spectrum in combination with the other jet bins. The $p_T^{Hjj}$ boundary here is kept as a sub-bin boundary.

– The $\boxed{m_{jj} > 350\,\text{GeV}}$ bin contains only a small fraction of the total $gg \to H$ cross section, which however constitutes the main background to VBF production. Hence it uses the same splitting as the corresponding high-$m_{jj}$ VBF bin in Section 4.2 with boundaries defined at $m_{jj} = 700, 1000$, and $1500\,\text{GeV}$. Currently, the $m_{jj} = 700\,\text{GeV}$ boundary defines an explicit bin separation, while the higher $m_{jj}$ boundaries are kept as sub-bins. The $p_T^{Hjj}$ boundary is an explicit bin separation.

## 4.2 Electroweak $qqH$ production (VBF + hadronic $VH$)

The VBF template process is defined more precisely as electroweak $qqH$ production. It includes the usual VBF topology and also the $pp \to V(\to q\bar{q})H$ topology with hadronic $V \to q\bar{q}$ decays. The two topologies lead to the same final state through the same interactions and therefore represent the $t$-channel and $s$-channel contributions to the same physical process. Hence, they can only be distinguished by enriching one or the other type of contribution via kinematic cuts, which is achieved by the STXS bins as described below.

The changes compared to the previous stage 1.0 is the treatment of the BSM sensitive high-$p_T$ region (which is now split out after the $m_{jj}$ separation), a more fine-grained $m_{jj}$ binning along with dropping the additional $|\Delta\eta_{jj}|$ cut, and the separation of the previous "Rest"-bin, which contained a combination of different jet topologies and kinematic regions. These are now separated to allow for an easier treatment, in particular for the estimation of theory uncertainties.

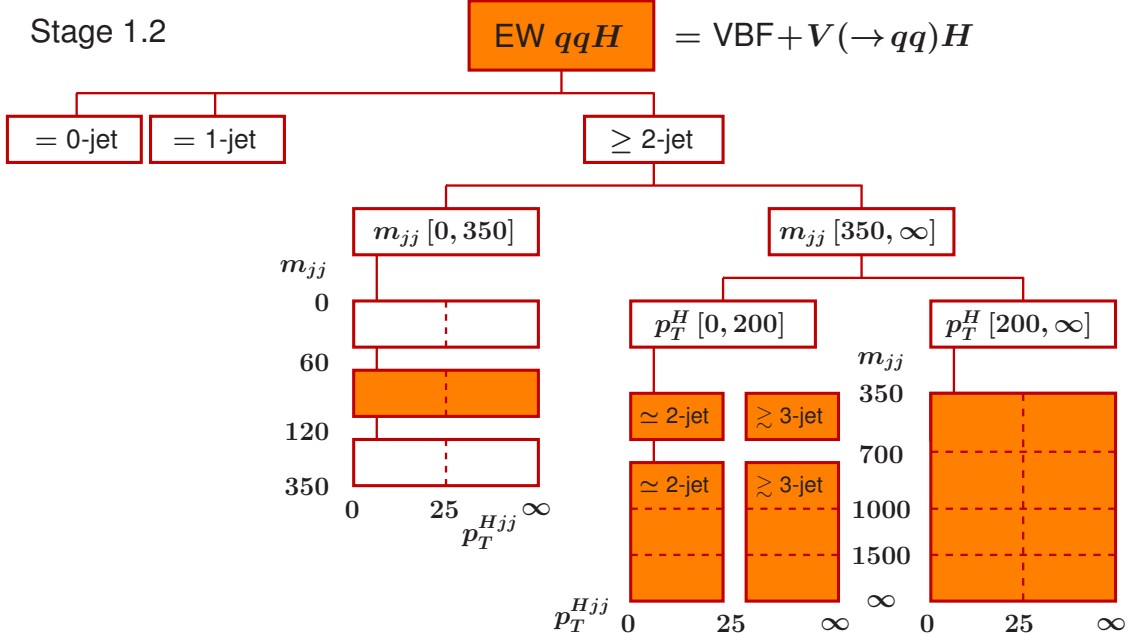

Figure 2: Stage 1.2 bins for electroweak $qqH$ production, VBF+$V(\to qq)H$. The stage 1.1 bins are identical.

The stage 1.2 bins, identical to the stage 1.1 ones, are depicted in Figure 2 and are described briefly in the following:

- The cross section is first split into $\boxed{\text{0-jet}}$, $\boxed{\text{1-jet}}$, and $\boxed{\geq \text{2-jet}}$ bins.

    - The $\boxed{\text{0-jet}}$ and $\boxed{\text{1-jet}}$ bins are very hard to access experimentally, and are likely to remain merged. It might be possible to get some sensitivity to the $\boxed{\text{1-jet}}$ bin using dedicated analyses. Previously they where included in the "Rest" bin.

    - The $\boxed{\geq \text{2-jet}}$ bin is the starting point for the remaining binning.

- The $\boxed{\geq \text{2-jet}}$ bin is split into low-$m_{jj}$ and high-$m_{jj}$ bins with $\boxed{m_{jj} < 350\,\text{GeV}}$ and $\boxed{m_{jj} > 350\,\text{GeV}}$, respectively.

    - The $\boxed{m_{jj} < 350\,\text{GeV}}$ bin was previously part of the "Rest"-bin as well as the previous "VH"-bin. In this kinematic region, contributions from the actual VBF process of interest are still very hard to distinguish from the overwhelming gluon-fusion background. This bin is split into 3 $m_{jj}$ regions with cuts at $m_{jj} = 60$ and $120\,\text{GeV}$. The middle $\boxed{60\,\text{GeV} < m_{jj} < 120\,\text{GeV}}$ bin targets the hadronic $VH$-like production. (It is equivalent to the previous "VH"-bin.) In addition, sub-bin boundaries at $p_T^{Hjj} = 25\,\text{GeV}$ are defined for all $m_{jj}$ bins, primarily for consistency with the higher $m_{jj}$ bins.

    - The $\boxed{m_{jj} > 350\,\text{GeV}}$ bin targets the nominal VBF production process. Compared to the previous "VBF"-bin, the $m_{jj}$ threshold is slightly lowered (from previously $400\,\text{GeV}$) to capture more of the VBF signal. Furthermore, the $|\Delta\eta_{jj}|$ cut is dropped in favor of a more fine-grained $m_{jj}$ binning. This allows one to better account for the fact that different analyses can have substantially different sensitivities to different $m_{jj}$ regions. It also allows for an easier treatment of theory uncertainties, which can now be based on considering the one-dimensional $m_{jj}$ spectrum.

- The $\boxed{m_{jj} > 350\,\text{GeV}}$ bin is split into low-$p_T^H$ and high-$p_T^H$ bins with $\boxed{p_T^H < 200\,\text{GeV}}$ and $\boxed{p_T^H > 200\,\text{GeV}}$, respectively. The $p_T$ separation is moved inside the nominal VBF-like region to allow for a better isolation of the high-$p_T$ region of the actual VBF process. In addition, the $p_T$ variable is changed from the $p_T$ of the leading jet, which was used in stage 1.0, to the $p_T^H$ of the Higgs boson. The sensitivity to possible BSM effects at high $p_T$ is roughly similar for both variables. On the other hand, using $p_T^H$ has the important advantage that it better aligns with the use of $p_T^H$ in the $gg \to H$ bins, which have a large cross section and are hard to distinguish experimentally from the VBF process in this kinematic region. This allows for a much cleaner merging of corresponding bins across the VBF and $gg \to H$ processes if necessary.

    The bin has $m_{jj}$ boundaries defined at $m_{jj} = 700, 1000,$ and $1500\,\text{GeV}$. In addition, it has a bin boundary defined at $p_T^{Hjj} = 25\,\text{GeV}$, which provides a separation into 2-jet like and $\geq$ 3-jet like phase-space regions (as in stage 1.0), which is essential for the discrimination against the large gluon-fusion contributions.

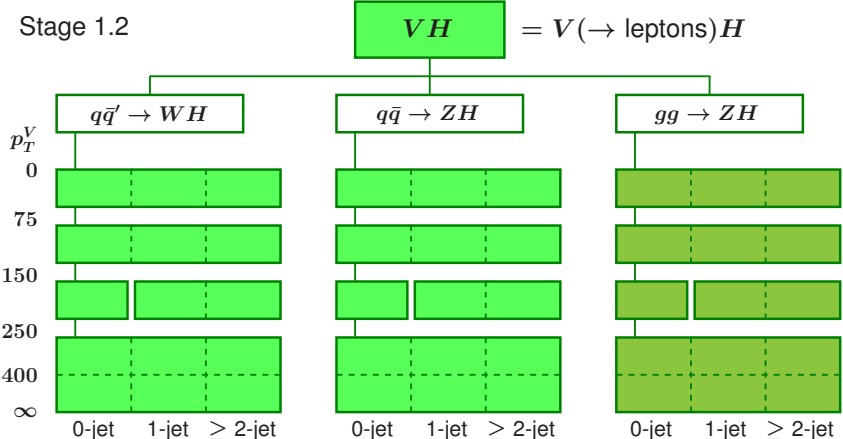

Figure 3: Stage 1.2 bins for $VH$ production, $V(\to \text{leptons})H$. The stage 1.1 bins are identical.

- The $\boxed{p_T^H < 200\,\text{GeV}}$ bin contains most of the (accessible) VBF signal. The $m_{jj} = 700\,\text{GeV}$ boundary is an explicit bin separation, while the higher $m_{jj}$ boundaries are kept as sub-bins at the current stage. The $p_T^{Hjj}$ boundary is an explicit bin separation. Hence, a total of four bins are defined at this stage. However, explicitly splitting out the higher $m_{jj}$ bins at the defined $m_{jj}$ boundaries is encouraged if there is sufficient sensitivity from dedicated analyses to allow for their separate measurement.

- The $\boxed{p_T^H > 200\,\text{GeV}}$ bin only contains a small fraction of the VBF signal and is therefore kept as a single bin at this stage, with all $m_{jj}$ boundaries and the $p_T^{Hjj}$ boundary kept as sub-bin boundaries.

## 4.3 Associated Higgs production (leptonic $VH$)

The $VH$ template process is defined as Higgs production in association with a leptonically decaying vector boson, $pp \to V(\to \text{leptons})H$. It is separated into the three underlying processes $q\bar{q}' \to W(\to \ell\bar{\nu})H$, $q\bar{q} \to Z(\to \ell\bar{\ell})H$, and $gg \to Z(\to \ell\bar{\ell})H$. The hadronic $VH$ processes $q\bar{q} \to V(\to q\bar{q})H$ are part of the electroweak $qqH$ template process (see Section 4.2). Similarly, the gluon-induced $gg \to Z(\to q\bar{q})H$ process is included as part of the $gg \to H$ template process (see Section 4.1), for which it represents an electroweak real-emission correction. The extensions in stage 1.1 are additional $p_T^V$ and jet-bin boundaries, and are fully backward compatible with the previous stage 1.0.

The stage 1.2 bins, identical to the stage 1.1 ones, are depicted in Figure 3 and are summarized in the following:

- The total cross section is first split into the subprocesses $\boxed{q\bar{q}' \to WH}$, $\boxed{q\bar{q} \to ZH}$ and $\boxed{gg \to ZH}$.

  - The $\boxed{q\bar{q}' \to WH}$ and $\boxed{q\bar{q} \to ZH}$ subprocesses are split into $p_T^V$ bins with boundaries at $p_T^V = 75, 150, 250$, and $400\,\text{GeV}$, where the $p_T^V = 400\,\text{GeV}$ bin boundary is kept as sub-bin at this stage. Compared to stage 1.0, the boundaries at $p_T^V = 75$ and $400\,\text{GeV}$ were added. This more fine grained $p_T^V$ binning better reflects the experimental sensitivity in the low $p_T^V$ range and also allows one to provide the theory uncertainties with sufficient detail.

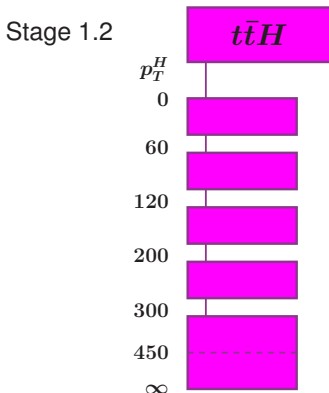

Figure 4: Stage 1.2 bins for $t\bar{t}H$ production. In stage 1.1 this production mode is not binned.

- Exactly the same binning as for $\boxed{q\bar{q} \to ZH}$ is now used for $\boxed{gg \to ZH}$. This allows for a more consistent merging of individual bins across the two subprocesses, which at present are hard to separate experimentally. In addition, it facilitates a better treatment of the sizeable theory uncertainties for $gg \to ZH$.

- As in stage 1.0, the $\boxed{150\,\text{GeV} < p_T^V < 250\,\text{GeV}}$ bin is split explicitly into $\boxed{0\text{-jet}}$ and $\boxed{\geq 1\text{-jet}}$ bins. Stage 1.1 and 1.2 now also add 0-jet, 1-jet, $\geq$ 2-jet sub-bins in all $p_T^V$ bins to allow for a more fine-grained estimate of theory uncertainties.

## 4.4 $t\bar{t}H$ production

The $t\bar{t}H$ template process is defined as Higgs production in association with a $t\bar{t}$ pair. While in stage 1.1 this process is not split into bins, stage 1.2 incorporates five bins in $p_T^H$ with boundaries at $p_T^H = 60, 120, 200$, and $300\,\text{GeV}$. An additional sub-bin boundary at $p_T^H > 450\,\text{GeV}$ is also introduced to enhance sensitivity to the boosted topology. The stage 1.2 bins for the $t\bar{t}H$ production mode are depicted in Figure 4. The observable $p_T^H$ is chosen for the binning, as it has good sensitivity to potential BSM effects, while being simpler than other, more complex observables. In particular, it has the key advantage that it does not require one to define a top quark or top jet as a truth-level object.

## 4.5 Other production modes

### 4.5.1 $b\bar{b} \to H$ production

So far it is not possible in experimental analyses to separate the $b\bar{b} \to H$ process from the by far larger $gg \to H$ process, and this is likely to remain the case in the near future. For this reason, the two processes are currently merged and $b\bar{b} \to H$ should hence be used with a binning that follows that of $gg \to H$ as needed by each analysis.

### 4.5.2 $tH$ production

Due to the low experimental sensitivity using the LHC Run 2 dataset to measure Higgs boson production in association with a single top quark, comprising $tHq$ and $tWH$ production modes, these processes are treated commmonly as $tH$ production and are not split into bins in stages 1.1. and 1.2. The split of these processes is planned for a future stage.

# 5 Conclusions

We have presented the complete definitions of Simplified Template Cross Sections in stage 1.2 and its predecessor stage 1.1. These have been used for the measurements based on the full Run 2 datasets by the ATLAS and CMS experiments. Compared to the previous stage 1.0, several refinements and extensions have been introduced, in particular for the VBF process. For stage 1.2, measurements in the $t\bar{t}H$ production mode are divided into bins.

A new feature in stage 1.1 and stage 1.2 is the introduction of sub-bin boundaries. Their purpose is to allow for an improved treatment of residual theory uncertainties in the signal distributions and their propagation to the measured parameters. For this reason, the full granularity including sub-bins is higher than the experimental sensitivity with the full Run 2 datasets. The sub-bin boundaries should be considered as possible boundaries for splitting bins, allowing for a smoother evolution of the binning in the future. The final bin splitting or merging should be optimized by the experiments based on the available statistics at a given time.

# Acknowledgments

**Funding information**  This work was supported in part by the French Agence Nationale de la Recherche under the project "PhotonPortal" (ANR-16-CE31-0016); the US National Science Foundation award 1806579; Imperial College London and the Science and Technology Facilities Council, UK; and the Deutsche Forschungsgemeinschaft (DFG) under Germany's Excellence Strategy – EXC 2121 "Quantum Universe" – 390833306.

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
