# Peer review of "Simplified Template Cross Sections -- Stage 1.1 and 1.2"

_SciPost Physics Community Reports, doi:SciPost Phys. Comm. Rep. 15 (2026)_

## Round 2 · Referee Report · Anonymous (Referee 1) · 2025-7-16

Report
Recommendation
Publish (surpasses expectations and criteria for this Journal; among top 10%)

Author: Frank Tackmann on 2025-11-04 [id 5986]
(in reply to Report 2 on 2025-08-13)We thank the referee for their suggestions and we apologize for the delay in response.
In principle, we agree of course that it would be extremely useful to have SM state-of-the art reference predictions for all the STXS bins available. However, such a compilation of results is unfortunately still not available at present. Providing one is clearly well beyond the scope of this paper., as it is even more involved than the compilation of the total production cross sections provided by the LHC Higgs WG, which already requires a concerted community-wide effort. We of course hope that at some point in the future this would become a reality in the context of the LHC Higgs WG or otherwise.
Regarding the old vs. new name of the WG: The place it appears in the introduction is in a historical context. The paper originally appeared when stage 1.1 was agreed upon which indeed happened before the name change. We therefore decided to keep the old name here to reflect the correct history.

---

## Round 2 · Referee Report · Anonymous (Referee 2) · 2025-8-13

Report
I find difficult to suggest any point where the article could be improved. If anything, I could suggest that, in the same way the LHC Higgs WG provides the SM predictions for the different Higgs cross sections (see https://twiki.cern.ch/twiki/bin/view/LHCPhysics/LHCHWG136TeVxsec_extrap), it would be useful to have available, either in the article or in some ancilliary material, the SM predictions and uncertainties for the different STXS 1.2 bins (both at 13.6 TeV for LHC Run 3 and 14 TeV for HL-LHC projections).
Other than that, and as a very minor thing, I noticed the authors still refer in the introduction to the LHC Higgs Working Group by its old name (LHC Higgs Cross Section Working Group), so they may want to update that.
I am happy to recommend this document for its publication in SciPost.
Recommendation
Publish (surpasses expectations and criteria for this Journal; among top 10%)

---

## Editorial Decision

published